

# A systematic literature review on the usability of mobile applications for visually impaired users

Muna Al-Razgan[1], Sarah Almoaiqel[1], Nuha Alrajhi[2], Alyah Alhumegani[1], Abeer Alshehri[2], Bashayr Alnefaie[1], Raghad AlKhamiss[1] and Shahad Rushdi[1]

[1] King Saud University, Riyadh, Saudi Arabia
[2] Imam Muhammad Ibn Saud University, Riyadh, Saudi Arabia

## ABSTRACT

Interacting with mobile applications can often be challenging for people with visual impairments due to the poor usability of some mobile applications. The goal of this paper is to provide an overview of the developments on usability of mobile applications for people with visual impairments based on recent advances in research and application development. This overview is important to guide decision-making for researchers and provide a synthesis of available evidence and indicate in which direction it is worthwhile to prompt further research. We performed a systematic literature review on the usability of mobile applications for people with visual impairments. A deep analysis following the Preferred Reporting Items for SLRs and Meta-Analyses (PRISMA) guidelines was performed to produce a set of relevant papers in the field. We first identified 932 papers published within the last six years. After screening the papers and employing a snowballing technique, we identified 60 studies that were then classified into seven themes: accessibility, daily activities, assistive devices, navigation, screen division layout, and audio guidance. The studies were then analyzed to answer the proposed research questions in order to illustrate the different trends, themes, and evaluation results of various mobile applications developed in the last six years. Using this overview as a foundation, future directions for research in the field of usability for the visually impaired (UVI) are highlighted.

## INTRODUCTION

The era of mobile devices and applications has begun. With the widespread use of mobile applications, designers and developers need to consider all types of users and develop applications for their different needs. One notable group of users is people with visual impairments. According to the World Health Organization, there are approximately 285 million people with visual impairments worldwide (*World Health Organization, 2020*). This is a huge number to keep in mind while developing new mobile applications.

People with visual impairments have urged more attention from the tech community to provide them with the assistive technologies they need (*Khan & Khusro, 2021*). Small tasks that we do daily, such as picking out outfits or even moving from one room to another,

Corresponding author
Sarah Almoaiqel,
salmoaiqel@ksu.edu.sa

could be challenging for such individuals. Thus, leveraging technology to assist with such tasks can be life changing. Besides, increasing the usability of applications and developing dedicated ones tailored to their needs is essential. The usability of an application refers to its efficiency in terms of the time and effort required to perform a task, its effectiveness in performing said tasks, and its users' satisfaction (*Ferreira et al., 2020*). Researchers have been studying this field intensively and proposing different solutions to improve the usability of applications for people with visual impairments.

This paper provides a systematic literature review (SLR) on the usability of mobile applications for people with visual impairments. The study aims to find discussions of usability issues related to people with visual impairments in recent studies and how they were solved using mobile applications. By reviewing published works from the last six years, this SLR aims to update readers on the newest trends, limitations of current research, and future directions in the research field of usability for the visually impaired (UVI).

This SLR can be of great benefit to researchers aiming to become involved in UVI research and could provide the basis for new work to be developed, consequently improving the quality of life for the visually impaired. This review differs from previous review studies (*i.e., Khan & Khusro, 2021*) because we classified the studies into themes in order to better evaluate and synthesize the studies and provide clear directions for future work. The following themes were chosen based on the issues addressed in the reviewed papers: "Assistive Devices," "Navigation," "Accessibility," "Daily Activities," "Audio Guidance," and "Gestures." Figure 1 illustrates the percentage of papers classified in each theme.

The remainder of this paper is organized as follows: the next section specifies the methodology, following this, the results section illustrates the results of the data collection, the discussion section consists of the research questions with their answers and the limitations and potential directions for future work, and the final section summarizes this paper's main findings and contribution.

## SURVEY METHODOLOGY

This systematic literature review used the Meta-Analyses (PRISMA, 2009) guidelines to produce a set of relevant papers in the field. This SLR was undertaken to address the research questions described below. A deep analysis was performed based on a group of studies; the most relevant studies were documented, and the research questions were addressed.

### A. Research questions

The research questions addressed by this study are presented in Table 1 with descriptions and the motivations behind them.

### B. Search strategy

This review analysed and synthesised studies on usability for the visually impaired from a user perspective following a systematic approach. As proposed by *Tanfield, Denyer & Smart (2003)*, the study followed a three-stage approach to ensure that the findings were both reliable and valid. These stages were planning the review, conducting the review by

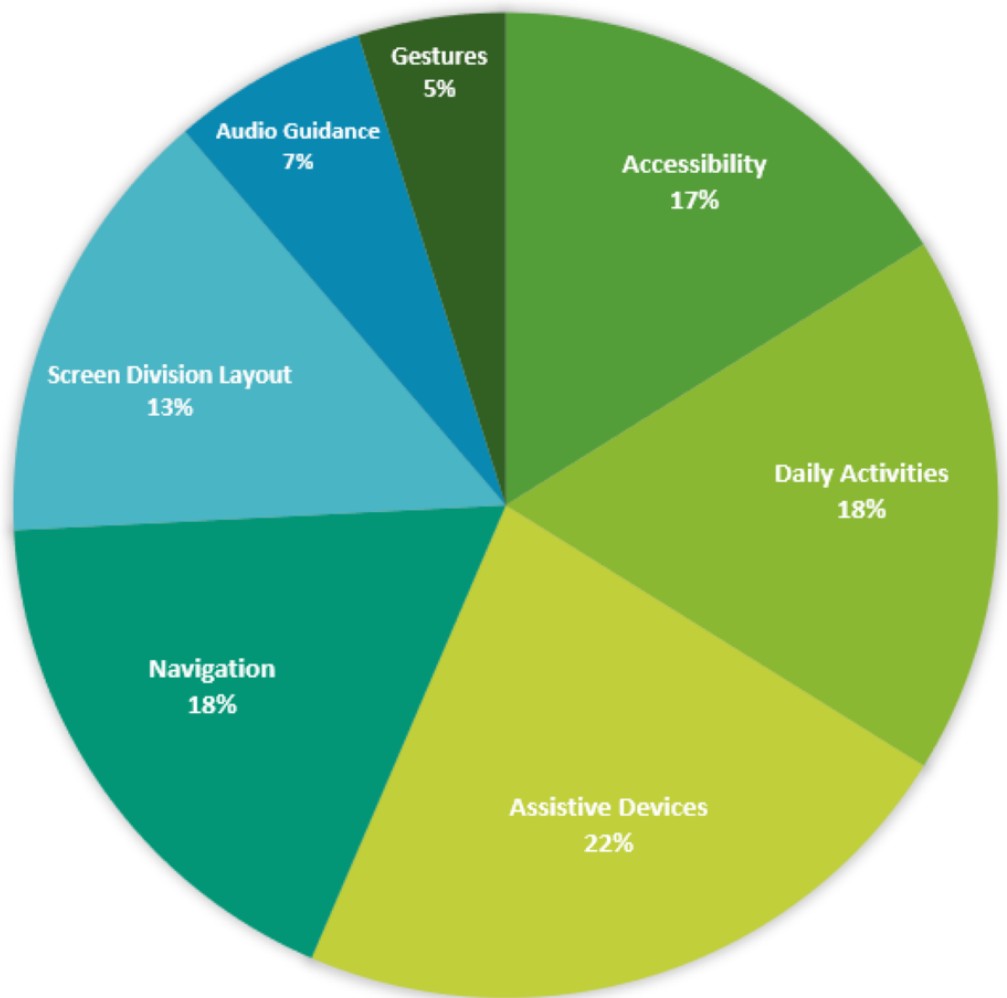

**Figure 1**   **Percentages of classification themes.**

analysing papers, and reporting emerging themes and recommendations. These stages will be discussed further in the following section.

### 1. Planning stage

The planning stage of this review included defining data sources and the search string protocol as well as inclusion and exclusion criteria.

*Data sources.*  We aimed to use two types of data sources: digital libraries and search engines. The search process was manually conducted by searching through databases. The selected databases and digital libraries are as follows:

- ACM Library
- IEEE Xplore
- ScienceDirect

**Table 1 Research questions.**

| Research question | Description and motivation |
|---|---|
| **RQ1:** What existing UVI issues did authors try to solve with mobile devices?; | The issues and proposed solutions will be of great significance for researchers as well as developers, providing a deeper understanding of whether a specific problem was addressed in the literature and what the proposed solutions were. |
| **RQ2:** What is the role of mobile devices in solving those issues? | Being able to identify the role of mobile devices in assisting visually impaired people in their daily lives will help improve their usability and provide a basis for future applications to be developed to improve quality of life for the visually impaired. |
| **RQ3:** What are the publication trends on the usability of mobile applications among the visually impaired? | After answering this question, it will become easier to classify the current existing work and the available application themes for the visually impaired. |
| **RQ4:** What are the current research limitations and future research directions regarding usability among the visually impaired? | This will help guide future research and open doors for new development. |
| **RQ5:** What is the focus of research on usability for visually impaired people, and what are the research outcomes in the ;studies reviewed? | Answering this question, will enable us to address the current focus of studies and the available ways to collect data. |
| **RQ6:** What evaluation methods were used in the studies on usability for visually impaired people that were reviewed? | This evaluation will help future researchers choose the most suitable methods according to the nature of their studies. |

- SpringerLink
- ISI Web of Knowledge
- Scopus.

The selected search engines were as follows:

- DBLP (Computer Science Bibliography Website)
- Google Scholar
- Microsoft Academic

*Search string.* The above databases were initially searched using the following keyword protocol: (''Usability'' AND (''visual impaired'' OR ''visually impaired'' OR ''blind'' OR ''impairment'') AND ''mobile''). However, in order to generate a more powerful search string, the Network Analysis Interface for Literature Studies (NAILS) project was used. NAILS is an automated tool for literature analysis. Its main function is to perform statistical and social network analysis (SNA) on citation data (*Knutas et al., 2015*). In this study, it was used to check the most important work in the relevant fields as shown in Fig. 2.

NAILS produced a report displaying the most important authors, publications, and keywords and listed the references cited most often in the analysed papers (*Knutas et al., 2015*) . The new search string was generated after using the NAILS project as follows: (''Usability'' OR ''usability model'' OR ''usability dimension'' OR ''Usability evaluation model'' OR ''Usability evaluation dimension'') AND (''mobile'' OR ''Smartphone'') AND (''Visually impaired'' OR ''Visual impairment'' OR ''Blind'' OR ''Low vision'' OR ''Blindness'').

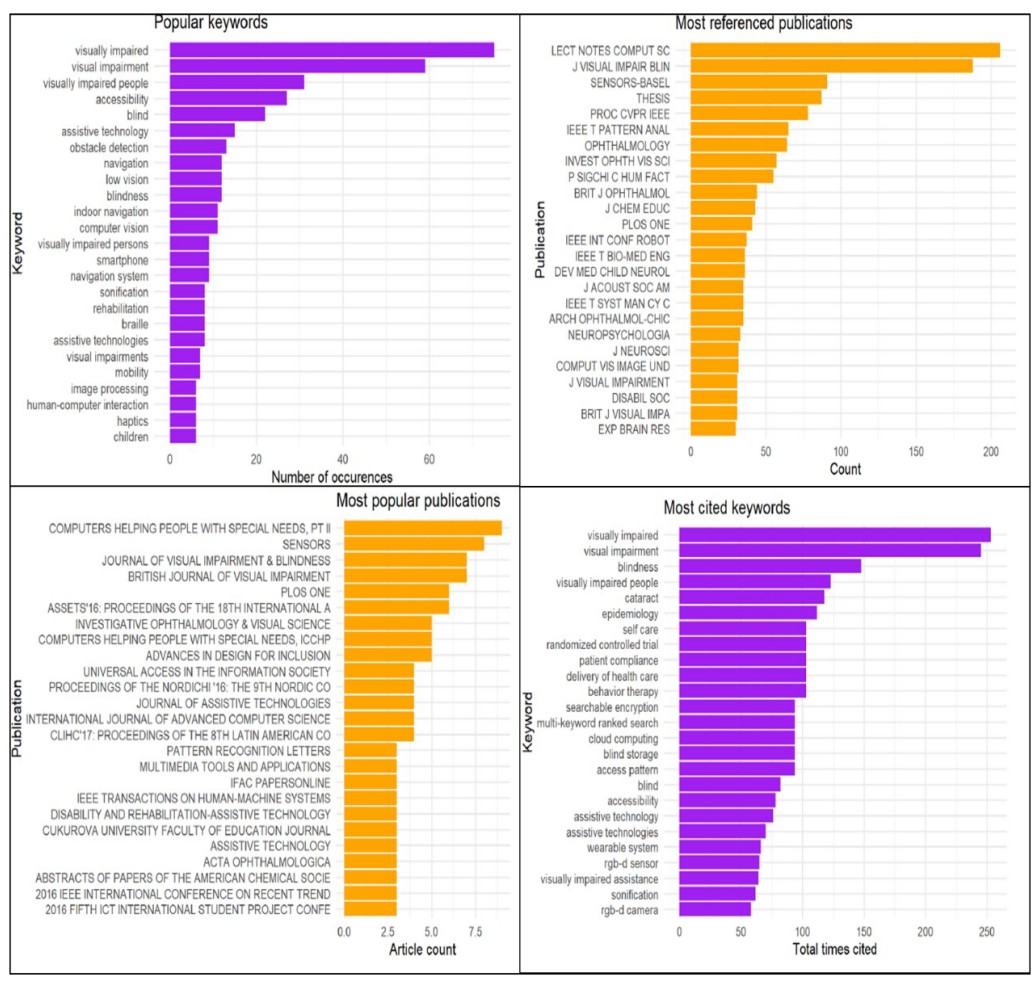

**Figure 2** NAILS output sample.

*Inclusion and exclusion criteria.* To be included in this systematic review, each study had to meet the following screening criteria:

- The study must have been published between 2015 and 2020.
- The study must be relevant to the main topic (Usability of Mobile Applications for Visually Impaired Users).
- The study must be a full-length paper.
- The study must be written in English because any to consider any other languages, the research team will need to use the keywords of this language in this topic and deal with search engines using that language to extract all studies related to our topic to form an SLR with a comprehensive view of the selected languages. Therefore, the research team preferred to focus on studies in English to narrow the scope of this SLR.

A research study was excluded if it did not meet one or more items of the criteria.

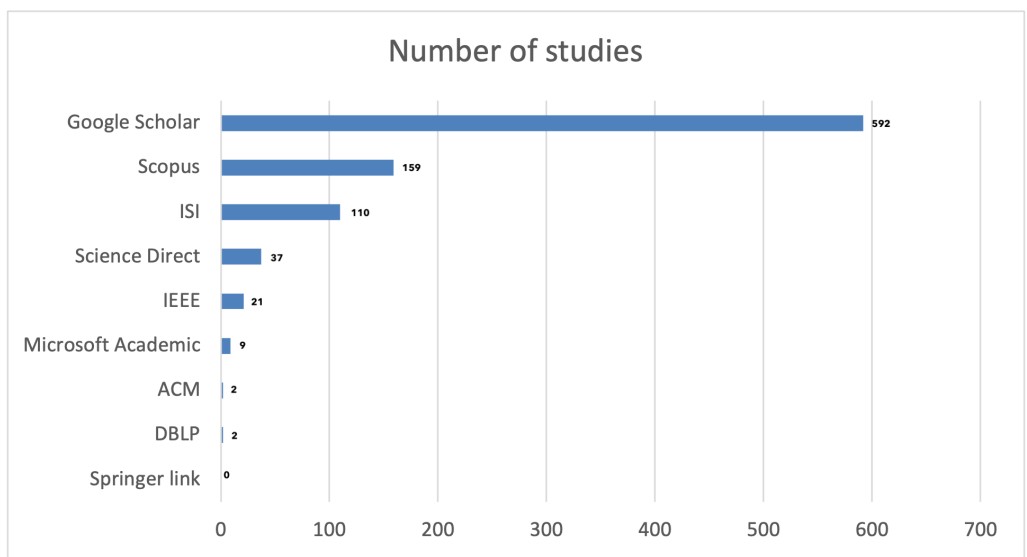

**Figure 3** **Number of papers per database.**

## 2. Conducting stage

The conducting stage of the review involved a systematic search based on relevant search terms. This consisted of three substages: exporting citations, importing citations into Mendeley, and importing citations into Rayyan.

*Exporting citations.* First, in exporting the citations and conducting the search through the mentioned databases, a total of 932 studies were found. The numbers are illustrated in Fig. 3 below. The highest number of papers was found in Google Scholar, followed by Scopus, ISI Web of Knowledge, ScienceDirect, IEEE Xplore, Microsoft Academic, and DBLP and ACM Library with two studies each. Finally, SpringerLink did not have any studies that met the inclusion criteria.

The chance of encountering duplicate studies was determined to be high. Therefore, importing citations into Mendeley was necessary in order to eliminate the duplicates.

*Importing citations into mendeley.* Mendeley is an open-source reference and citation manager. It can highlight paragraphs and sentences, and it can also list automatic references on the end page. Introducing the use of Mendeley is also expected to avoid duplicates in academic writing, especially for systematic literature reviews (*Basri & Patak, 2015*). Hence, in the next step, the 932 studies were imported into Mendeley, and each study's title and abstract were screened independently for eligibility. A total of 187 duplicate studies were excluded. 745 total studies remained after the first elimination process. The search stages are shown in Fig. 4 below.

*Importing citations into rayyan.* Rayyan QCRI is a free web and mobile application that helps expedite the initial screening of both abstracts and titles through a semi-automated process while incorporating a high level of usability. Its main benefit is to speed up the

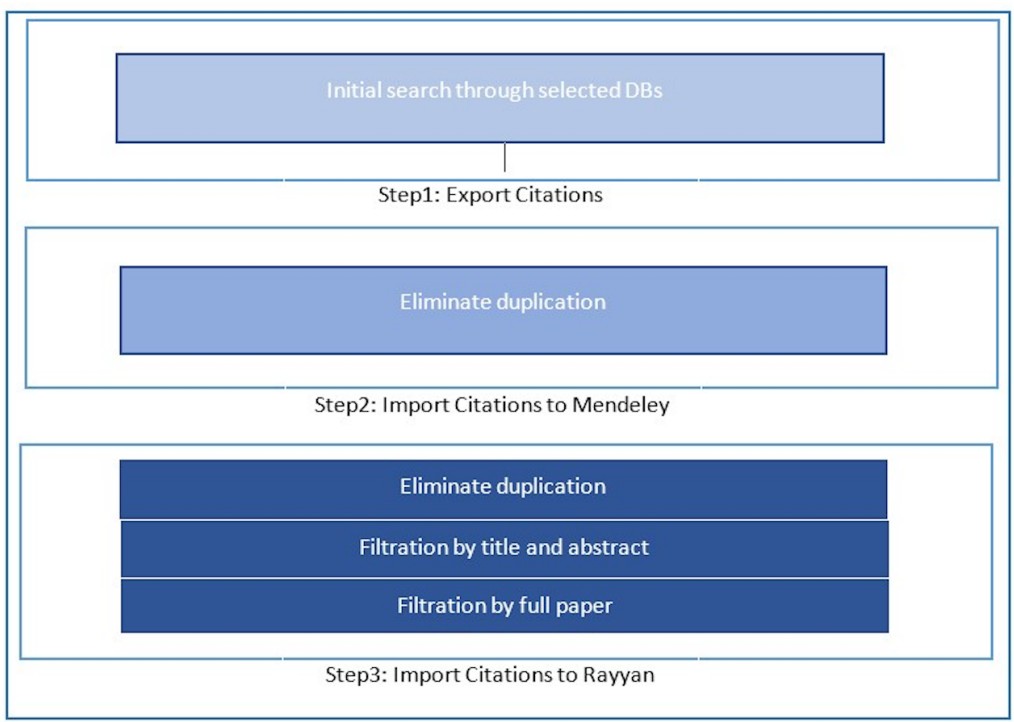

**Figure 4** Search stages.

most tedious part of the systematic literature review process: selecting studies for inclusion in the review (*Ouzzani et al., 2016*). Therefore, for the last step, another import was done using Rayyan to check for duplications a final time. Using Rayyan, a total of 124 duplicate studies were found, resulting in a total of 621 studies. Using Rayyan, a two-step filtration was conducted to guarantee that the papers have met the inclusion criteria of this SLR. After filtering based on the abstracts, 564 papers did not meet the inclusion criteria. At this stage, 57 studies remained. The second step of filtration eliminated 11 more studies by reading the full papers; two studies were not written in the English language, and nine were inaccessible.

*Snowballing.* Snowballing is an emerging technique used to conduct systematic literature reviews that are considered both efficient and reliable using simple procedures. The procedure for snowballing consisted of three phases in each cycle. The first phase is refining the start set, the second phase is backward snowballing, and the third is forward snowballing. The first step, forming the start set, is basically identifying relevant papers that can have a high potential of satisfying the criteria and research question. Backward snowballing was conducted using the reference list to identify new papers to include. It shall start by going through the reference list and excluding papers that do not fulfill the basic criteria; the rest that fulfil criteria shall be added to the SLR. Forward snowballing refers to identifying new papers based on those papers that cited the paper being examined (*Juneja & Kaur, 2019*). Hence, in order to be sure that we concluded all related studies after

we got the 46 papers, a snowballing step was essential. Forward and backward snowballing were conducted. Each of the 46 studies was examined by checking their references to take a look at any possible addition of sources and examining all papers that cited this study. The snowballing activity added some 38 studies, but after full reading, it became 33 that matched the inclusion criteria. A total of 79 studies were identified through this process.

*Quality assessment.* A systematic literature review's quality is determined by the content of the papers included in the review. As a result, it is important to evaluate the papers carefully (*Zhou et al., 2015*). Many influential scales exist in the software engineering field for evaluating the validity of individual primary studies and grading the overall intensity of the body of proof. Hence, we adapted the comprehensive guidelines specified by Kitchenhand and Charters (*Keele, 2007*), and the quasi-gold standard (QGS) (*Keele, 2007*) was used to establish the quest technique, where a robust search strategy for enhancing the validity and reliability of a SLR's search process is devised using the QGS. By applying this technique, our quality assessment questions were focused and aligned with the research questions mentioned earlier.

In our last step, we had to verify the papers' eligibility; we conducted a quality check for each of the 79 studies. For quality assessment, we considered whether the paper answered the following questions:

QA1: Is the research aim clearly stated in the research?

QA2: Does the research contain a usability dimension or techniques for mobile applications for people with visual impairments?

QA3: Is there an existing issue with mobile applications for people with visual impairments that the author is trying to solve?

QA4: Is the research focused on mobile application solutions?

After discussing the quality assessment questions and attempting to find an answer in each paper, we agreed to score each study per question. If the study answers a question, it will be given 2 points; if it only partially answers a question, it will be given 1 point; and if there is no answer for a given question in the study, it will have 0 points.

The next step was to calculate the weight of each study. If the total weight was higher or equal to four points, the paper was accepted in the SLR; if not, the paper was discarded since it did not reach the desired quality level. Figure 5 below illustrates the quality assessment process. After applying the quality assessment, 39 papers were rejected since they received less than four points, which resulted in a final tally of 60 papers.

To summarize, this review was conducted according to the Preferred Reporting Items for SLRs and Meta-Analyses (PRISMA) (*Liberati et al., 2009*). The PRISMA diagram shown in Fig. 6 illustrates all systematic literature processes used in this study.

### 3. Analysing stage

All researchers involved in this SLR collected the data. The papers were distributed equally between them, and each researcher read each paper completely to determine its topic, extract the paper's limitations and future work, write a quick summary about it, and record this information in an Excel spreadsheet.

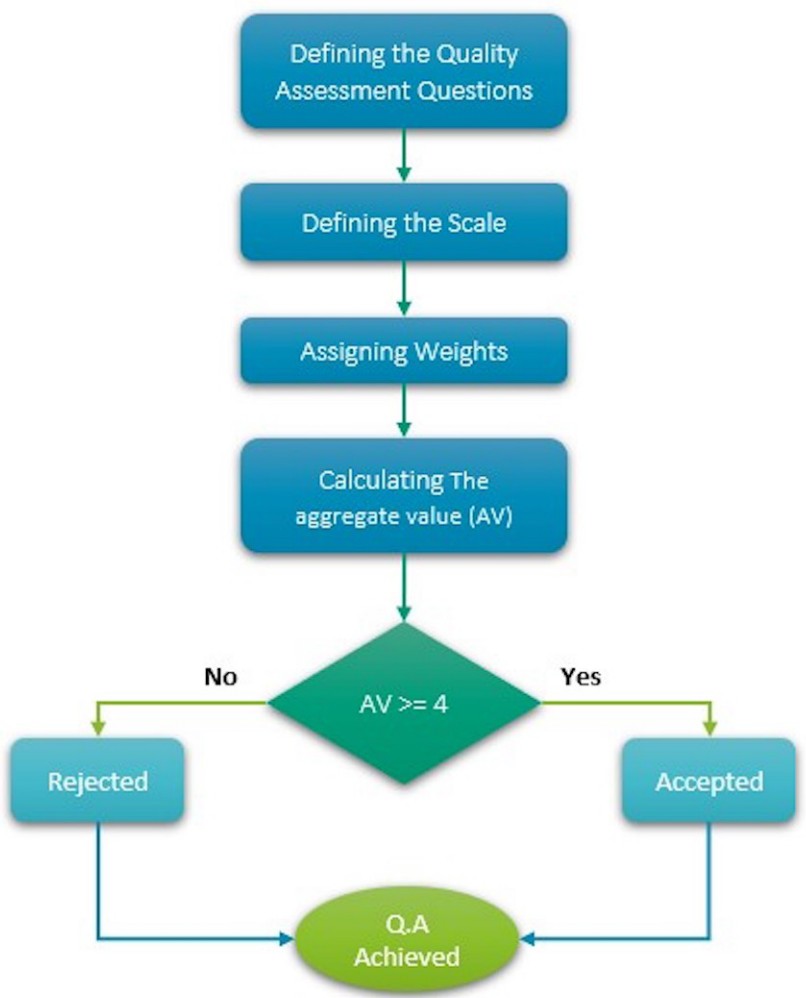

**Figure 5** **Quality assessment process.**

## RESULTS

All researchers worked intensively on this systematic literature review. After completing the previously mentioned steps, the papers were divided among all the researchers. Then, each researcher read their assigned papers completely and then classified them into themes according to the topic they covered. The researchers held several meetings to discuss and specify those themes. The themes were identified by the researchers based on the issues addressed in the reviewed papers. In the end, the researchers resulted in seven themes, as shown in Fig. 7 below. The references selected for each theme can be found in the Table A1. Afterwards, each researcher was assigned one theme to summarize its studies and report the results. In this section, we review the results.

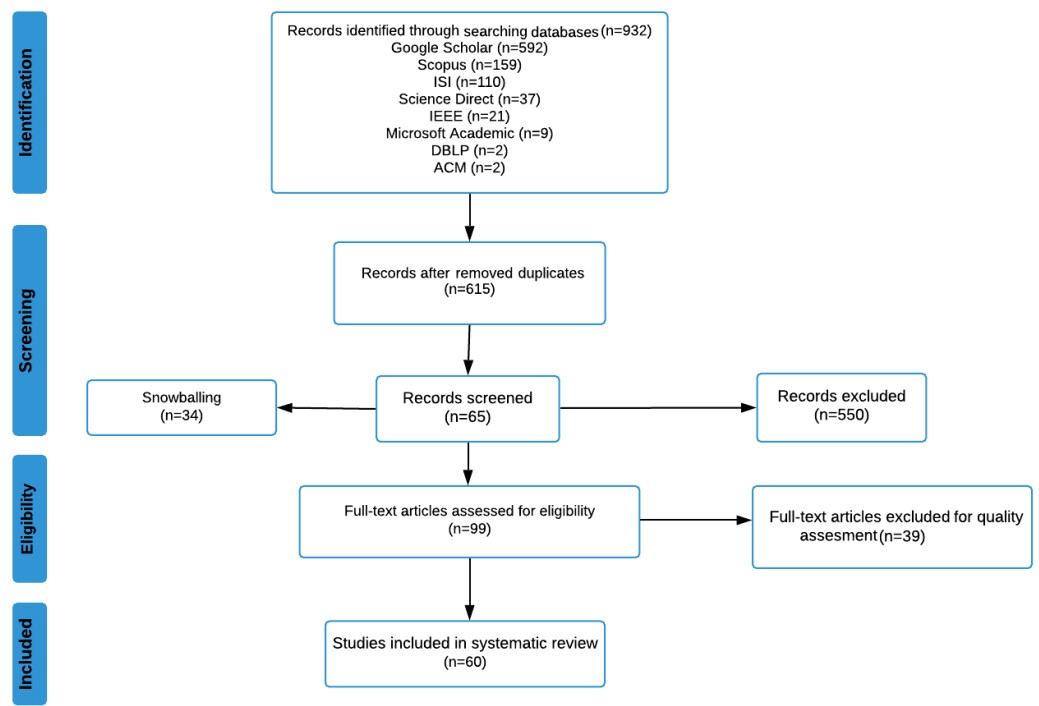

**Figure 6** PRISMA flow diagram.

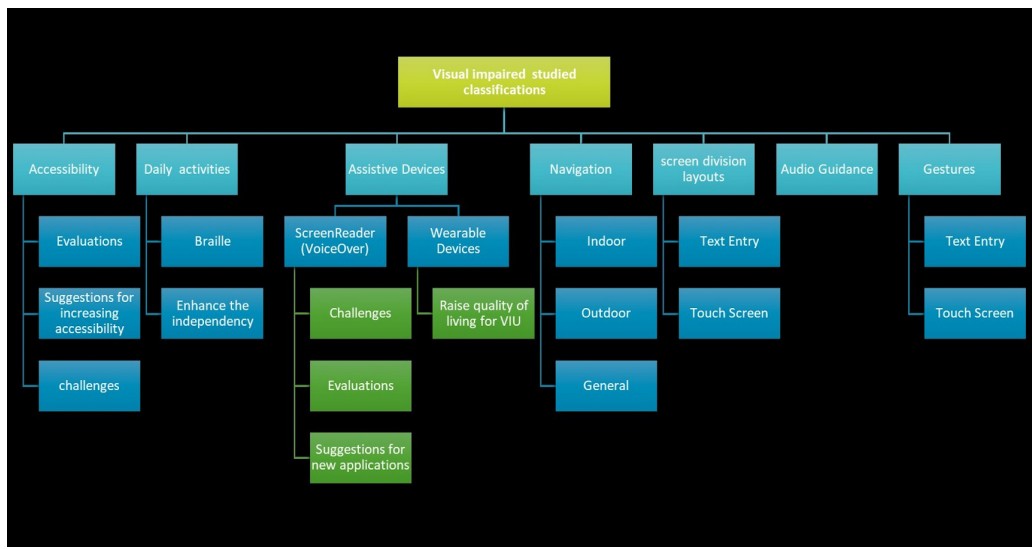

**Figure 7** Results of the SLR.

## A. Accessibility

Of a total of 60 studies, 10 focused on issues of accessibility. Accessibility is concerned with whether all users are able to have equivalent user experiences, regardless of abilities. Six studies, *Darvishy, Hutter & Frei (2019)*, *Morris et al. (2016)*, *Qureshi & Hooi-Ten Wong (2020)*, *Khan, Khusro & Alam (2018)*, *Paiva et al. (2020)*, and *Pereda, Murillo & Paz (2020)*, gave suggestions for increasing accessibility, (*Darvishy, Hutter & Frei, 2019*; *Morris et al., 2016*), gave some suggestions for making mobile map applications and Twitter accessible to visually impaired users, and (*Qureshi & Hooi-Ten Wong, 2020*; *Khan, Khusro & Alam, 2018*) focused on user interfaces and provided accessibility suggestions suitable for blind people. *Paiva et al. (2020)* and *Pereda, Murillo & Paz (2020)* proposed a set of heuristics to evaluate the accessibility of mobile applications. Two studies, *Khowaja et al. (2019)* and *Carvalho et al. (2018)*, focused on evaluating usability and accessibility issues on some mobile applications, comparing them, and identifying the number and types of problems that visually impaired users faced. *Aqle, Khowaja & Al-Thani (2020)* proposed a new web search interface designed for visually impaired users. One study, *McKay (2017)*, focused on accessibility challenges by applying usability tests on a hybrid mobile app with some visually impaired university students.

## B. Assistive devices

People with visual impairments have an essential need for assistive technology since they face many challenges when performing activities in daily life. Out of the 60 studies reviewed, 13 were related to assistive technology. The studies *Smaradottir, Martinez & Håland (2017)*, *Skulimowski et al. (2019)*, *Barbosa, Hayes & Wang, (2016)*, *Rosner & Perlman (2018)*, *Csapó et al. (2015)*, *Khan & Khusro (2020)*, *Sonth & Kallimani (2017)*, *Kim et al. (2016)*, *Vashistha et al. (2015)*; *Kameswaran et al. (2020)*, *Griffin-Shirley et al. (2017)*, and *Rahman, Anam & Yeasin (2017)* were related to screen readers (voiceovers). On the other hand, *Bharatia, Ambawane & Rane (2019)*, *Lewis et al. (2016)* were related to proposing an assistant device for the visually impaired. Of the studies related to screening readers, *Sonth & Kallimani, (2017)*, *Vashistha et al. (2015)*, *Khan & Khusro (2020) Lewis et al. (2016)* cited challenges faced by visually impaired users. *Barbosa, Hayes & Wang (2016)*, *Kim et al. (2016)*, *Rahman, Anam & Yeasin (2017)* suggested new applications, while *Smaradottir, Martinez & Håland (2017)*, *Rosner & Perlman (2018)*, *Csapó et al. (2015)* and *Griffin-Shirley et al. (2017)* evaluated current existing work. The studies *Bharatia, Ambawane & Rane (2019)*, *Lewis et al. (2016)* proposed using wearable devices to improve the quality of life for people with visual impairments.

## C. Daily activities

In recent years, people with visual impairments have used mobile applications to increase their independence in their daily activities and learning, especially those based on the braille method. We divide the daily activity section into braille-based applications and applications designed to enhance the independence of the visually impaired. Four studies, *Nahar, Sulaiman & Jaafar (2020)*, *Nahar, Jaafar & Sulaiman (2019)*, *Araújo et al. (2016)* and *Gokhale et al. (2017)*, implemented and evaluated the usability of mobile phone

applications that use braille to help visually impaired people in their daily lives. Seven studies, *Vitiello et al. (2018)*, *Kunaratana-Angkul, Wu & Shin-Renn (2020)*, *Ghidini et al. (2016)*, *Madrigal-Cadavid et al. (2019)*, *Marques, Carriço & Guerreiro (2015)*, *Oliveira et al. (2018)* and *Rodrigues et al. (2015)*, focused on building applications that enhance the independence and autonomy of people with visual impairments in their daily life activities.

## D. Screen division layout

People with visual impairments encounter various challenges in identifying and locating non-visual items on touch screen interfaces like phones and tablets. Incidents of accidentally touching a screen element and frequently following an incorrect pattern in attempting to access objects and screen artifacts hinder blind people from performing typical activities on smartphones (*Khusro et al., 2019*). In this review, 9 out of 60 studies discuss screen division layout: (*Khusro et al., 2019*; *Khan & Khusro, 2019*; *Grussenmeyer & Folmer, 2017*; *Palani et al., 2018*; *Leporini & Palmucci, 2018*) discuss touch screen (smartwatch tablets, mobile phones, and tablet) usability among people with visual impairments, while (*Cho & Kim, 2017*; *Alnfiai & Sampalli, 2016*; *Niazi et al., 2016*; *Alnfiai & Sampalli, 2019*) concern text entry methods that increase the usability of apps among visually impaired people. *Khusro et al. (2019)* provides a novel contribution to the literature regarding considerations that can be used as guidelines for designing a user-friendly and semantically enriched user interface for blind people. An experiment in *Cho & Kim (2017)* was conducted comparing the two-button mobile interface usability with the one-finger method and voiceover. *Leporini & Palmucci (2018)* gathered information on the interaction challenges faced by visually impaired people when answering questions on a mobile touch-screen device, investigated possible solutions to overcome the accessibility and usability challenges.

## E. Gestures

In total, 3 of 60 studies discuss gestures in usability. *Alnfiai & Sampalli (2017)* compared the performance of BrailleEnter, a gesture based input method to the Swift Braille keyboard, a method that requires finding the location of six buttons representing braille dot, while *Buzzi et al. (2017)* and *Smaradottir, Martinez & Haland (2017)* provide an analysis of gesture performance on touch screens among visually impaired people.

## F. Audio guidance

People with visual impairment primarily depend on audio guidance forms in their daily lives; accordingly, audio feedback helps guide them in their interaction with mobile applications.

Four studies discussed the use of audio guidance in different contexts: one in navigation (*Gintner et al., 2017*), one in games (*Ara'ujo et al., 2017*), one in reading (*Sabab & Ashmafee, 2016*), and one in videos (*Façanha et al., 2016*). These studies were developed and evaluated based on usability and accessibility of the audio guidance for people with visual impairments and aimed to utilize mobile applications to increase the enjoyment and independence of such individuals.

### G. Navigation

Navigation is a common issue that visually impaired people face. Indoor navigation is widely discussed in the literature. *Nair et al. (2020)*, *Al-Khalifa & Al-Razgan (2016)* and *De Borba Campos et al. (2015)* discuss how we can develop indoor navigation applications for visually impaired people. Outdoor navigation is also common in the literature, as seen in *Darvishy et al. (2020)*, *Hossain, Qaiduzzaman & Rahman (2020)*, *Long et al. (2016)*, *Prerana et al. (2019)* and *Bandukda et al. (2020)*. For example, in *Darvishy et al. (2020)*, Touch Explorer, an accessible digital map application, was presented to alleviate many of the problems faced by people with visual impairments while using highly visually oriented digital maps. Primarily, it focused on using non-visual output modalities like voice output, everyday sound, and vibration feedback. Issues with navigation applications were also presented in *Maly et al. (2015)*. *Kameswaran et al. (2020)* discussed commonly used technologies in navigation applications for blind people and highlighted the importance of using complementary technologies to convey information through different modalities to enhance the navigation experience. Interactive sonification of images for navigation has also been shown in *Skulimowski et al. (2019)*.

## DISCUSSION

In this section, the research questions are addressed in detail to clearly achieve the research objective. Also, a detailed overview of each theme will be mentioned below.

### Answers to the research questions

This section will answer the research question proposed:

*RQ1: What existing UVI issues did authors try to solve with mobile devices?*
Mobile applications can help people with visual impairments in their daily activities, such as navigation and writing. Additionally, mobile devices may be used for entertainment purposes. However, people with visual impairments face various difficulties while performing text entry operations, text selection, and text manipulation on mobile applications (*Niazi et al., 2016*). Thus, the authors of the studies tried to increase touch screens' usability by producing prototypes or simple systems and doing usability testing to understand the UX of people with visual impairments.

*RQ2: What is the role of mobile devices in solving those issues?*
Mobile phones are widely used in modern society, especially among users with visual impairments; they are considered the most helpful tool for blind users to communicate with people worldwide (*Smaradottir, Martinez & Håland, 2017*). In addition, the technology of touch screen assistive technology enables speech interaction between blind people and mobile devices and permits the use of gestures to interact with a touch user interface. Assistive technology is vital in helping people living with disabilities perform actions or interact with systems (*Niazi et al., 2016*).

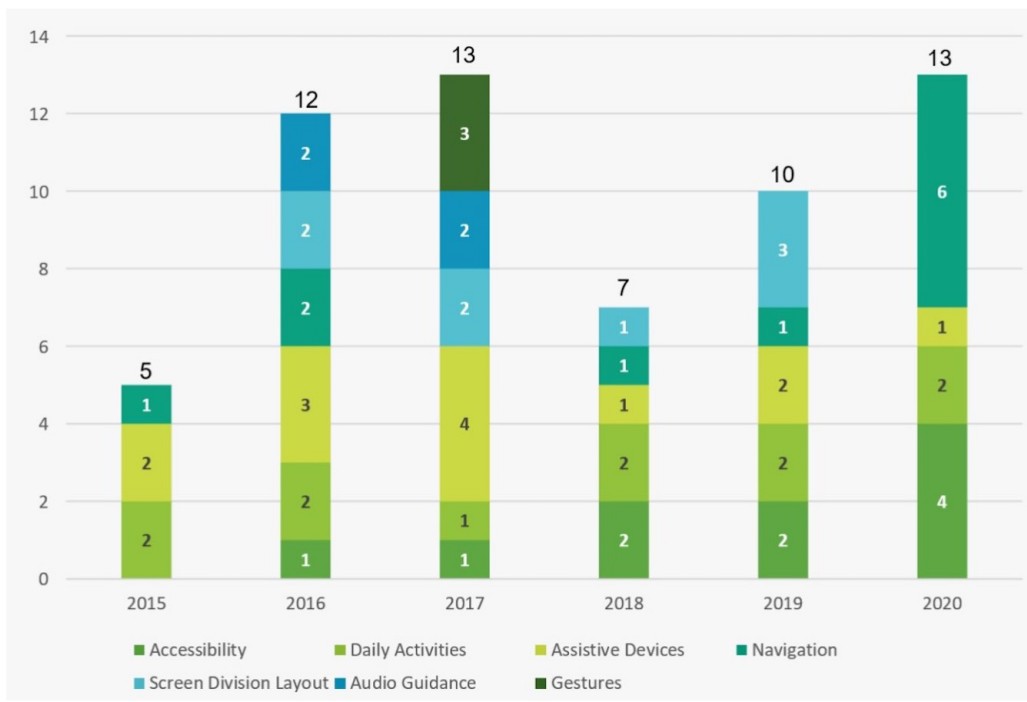

**Figure 8** **Publication trends over time.**

### RQ3: What are the publication trends on the usability of mobile applications among the visually impaired?

As shown in Fig. 8 below, research into mobile applications' usability for the visually impaired has increased in the last five years, with a slight dip in 2018. Looking at the most frequent themes, we find that "Assistive Devices" peaked in 2017, while "Navigation" and "Accessibility" increased significantly in 2020. On the other hand, we see that the prevalence of "Daily Activities" stayed stable throughout the research years. The term "Audio Guidance" appeared in 2016 and 2017 and has not appeared in the last three years. "Gestures" also appeared only in 2017. "Screen Layout Division" was present in the literature in the last five years and increased in 2019 but did not appear in 2020.

### RQ4: What are the current research limitations and future research directions regarding usability among the visually impaired?

We divide the answer to this question into two sections: first, we will discuss limitations; then, we will discuss future work for each proposed theme.

#### A. Limitations

Studies on the usability of mobile applications for visually impaired users in the literature have various limitations, and most of them were common among the studies. These limitations were divided into two groups. The first group concerns proposed applications; for example, *Rahman, Anam & Yeasin (2017)*, *Oliveira et al. (2018)* and *Madrigal-Cadavid et al. (2019)* faced issues regarding camera applications in mobile devices due to the

considerable effort needed for its usage and being heavily dependent on the availability of the internet. The other group of studies, *Rodrigues et al. (2015)*, *Leporini & Palmucci (2018)*, *Alnfiai & Sampalli (2016)*, and *Ara'ujo et al. (2017)*, have shown limitations in visually impaired users' inability to comprehend a graphical user interface. *Alnfiai & Sampalli (2017)* and *Alnfiai & Sampalli (2019)* evaluated new braille input methods and found that the traditional braille keyboard, where knowing the exact position of letters QWERTY is required, is limited in terms of usability compared to the new input methods. Most studies faced difficulties regarding the sample size and the fact that many of the participants were not actually blind or visually impaired but only blindfolded. This likely led to less accurate results, as blind or visually impaired people can provide more useful feedback as they experience different issues on a daily basis and are more ideal for this type of study. So, the need for a good sample of participants who actually have this disability is clear to allow for better evaluation results and more feedback and recommendations for future research.

**B. Future work**

A commonly discussed future work in the chosen literature is to increase the sample sizes of people with visual impairment and focus on various ages and geographical areas to generalize the studies. Table 2 summarizes suggestions for future work according to each theme. Those future directions could inspire new research in the field.

***RQ5: What is the focus of research on usability for visually impaired people, and what are the research outcomes in the studies reviewed?***

There are a total of 60 outcomes in this research. Of these, 40 involve suggestions to improve usability of mobile applications; four of them address problems that are faced by visually impaired people that reduce usability. Additionally, 16 of the outcomes are assessments of the usability of the prototype or model. Two of the results are recommendations to improve usability. Finally, the last two outcomes are hardware solutions that may help the visually impaired perform their daily activities. Figure 9 illustrates these numbers.

## OVERVIEW OF THE REVIEWED STUDIES

In the following subsections, we summarize all the selected studies based on the classified theme: accessibility, assistive devices, daily activities, screen division layout, gestures, audio guidance, and navigation. The essence of the studies will be determined, and their significance in the field will be explored.

## A. Accessibility

For designers dealing with mobile applications, it is critical to determine and fix accessibility issues in the application before it is delivered to the users (*Khowaja et al., 2019*). Accessibility refers to giving the users the same user experience regardless of ability. In *Khowaja et al. (2019)* and *Carvalho et al. (2018)*, the researchers focused on comparing the levels of accessibility and usability in different applications. They had a group of visually impaired users and a group of sighted users test out the applications to compare the number and type of problems they faced and determine which applications contained the most violations.

**Table 2  Theme-based future work.**

| Theme | Suggestions for future work | Sources |
|---|---|---|
| Accessibility | In terms of accessibility, in the future, there is potential in investigating concepts of how information will be introduced in a mobile application to increase accessibility VI users. In addition, future work directions include extending frameworks for visually complex or navigationally dense applications. Furthermore, emotion-based UI design may also be investigated to improve accessibility. Moreover, the optimization of GUI layouts and elements could be considered with a particular focus on gesture control systems and eye-tracking systems. | *Darvishy, Hutter & Frei (2019)*, *Khan, Khusro & Alam (2018)*, *Paiva et al. (2020)*, *Khowaja et al. (2019)* and *Carvalho et al. (2018)* |
| Assistive devices | In terms of assistive devices for people with visual impairments, there is potential for future direction in research into multimodal non-visual interaction (*e.g.*, sonification methods). Also, since there is very little available literature about how to go about prototype development and evaluation activities for assistive devices for users with no or little sight, it is important to investigate this to further develop the field. | *Skulimowski et al. (2019)*, *Bharatia, Ambawane & Rane (2019)*, *Csapó et al. (2015)* and *Rahman, Anam & Yeasin (2017)* |
| Daily activities | There is a need to evaluate the usability and accessibility of applications that aim to assist visually impaired users and improve restrictions in daily activities. | *Madrigal-Cadavid et al. (2019)*, *Oliveira et al. (2018)*, and *Rodrigues et al. (2015)* |
| Screen division layout | In terms of screen division layout, it is important to continuously seek to improve interfaces and provide feedback to make them more focused, more cohesive, and simpler to handle. A complete set of robust design guidelines ought to be created to provide a wide variety of non-visual applications with increased haptic access on a touchscreen device. | *Khusro et al., (2019)*, *Alnfiai & Sampalli (2019)*, *Palani et al. (2018)* and *Khan & Khusro (2019)* |
| Gestures | Gesture based interaction ought to be further investigated as it has the potential to greatly improve the way VI users communicate with mobile devices. Performance of gestures with various sizes of touch screens ought to be compared, as the size might have a significant effect on what is considered a usable gesture. | *Alnfiai & Sampalli (2017)* and *Buzzi et al. (2017)* |
| Navigation | Literature suggests that future work in the area of navigation should focus on eliminating busy graphical interfaces and relying on sounds. Studying more methods and integrating machine learning algorithms and hardware devices to provide accurate results regarding the identification of surrounding objects, and continuous updates for any upcoming obstacles, is also discussed in the literature as an important direction for future work. | *Darvishy, Hutter & Frei (2019)*, *Hossain, Qaiduzzaman & Rahman (2020)* and *Bandukda et al. (2020)* |
| Audio guidance | In terms of audio guidance, there is potential for future directions in expanding algorithms to provide audio guidance to assist in more situations. Authors also emphasise developing versions of the applications in more languages. | *Gintner et al. (2017)*, *Sabab & Ashmafee (2016)* and *Façanha et al. (2016)* |

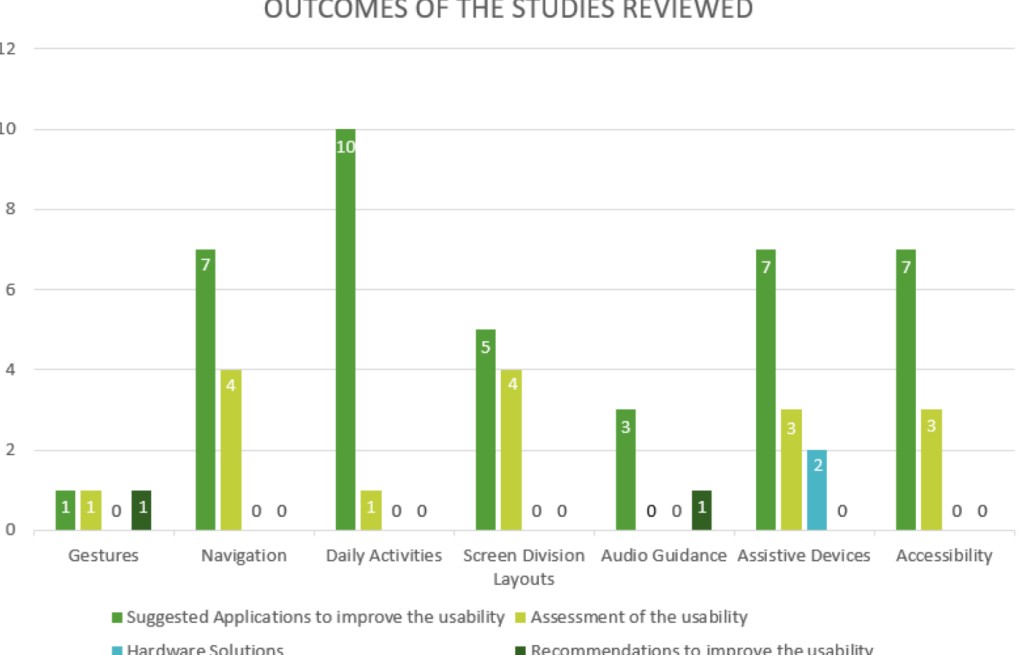

**Figure 9** Outcomes of studies.

Because people with visual impairments cannot be ignored in the development of mobile applications, many researchers have sought solutions for guaranteeing accessibility. For example, in *Qureshi & Hooi-Ten Wong (2020)*, the study contributed to producing a new, effective design for mobile applications based on the suggestions of people with visual impairments and with the help of two expert mobile application developers. In *Khan, Khusro & Alam (2018)*, an adaptive user interface model for visually impaired people was proposed and evaluated in an empirical study with 63 visually impaired people. In *Aqle, Khowaja & Al-Thani (2020)*, the researchers proposed a new web search interface for users with visual impairments that is based on discovering concepts through formal concept analysis (FCA). Users interact with the interface to collect concepts, which are then used as keywords to narrow the search results and target the web pages containing the desired information with minimal effort and time. The usability of the proposed search interface (InteractSE) was evaluated by experts in the field of HCI and accessibility, with a set of heuristics by Nielsen and a set of WCAG 2.0 guidelines.

In *Darvishy, Hutter & Frei (2019)*, the researchers proposed a solution for making mobile map applications accessible for people with blindness or visual impairment. They suggested replacing forests in the map with green color and birds' sound, replacing water with blue color and water sounds, replacing streets with grey color and vibration, and replacing buildings with yellow color and pronouncing the name of the building. The prototype showed that it was possible to explore a simple map through vibrations, sounds, and speech.

In *Morris et al. (2016)* the researchers utilized a multi-faceted technique to investigate how and why visually impaired individuals use Twitter and the difficulties they face in doing so. They noted that Twitter had become more image-heavy over time and that picture-based tweets are largely inaccessible to people with visual impairments. The researchers then made several suggestions for how Twitter could be amended to continue to be usable for people with visual impairments.

The researchers in *Paiva et al. (2020)* focused on how to evaluate proposed methods for ensuring the accessibility and usability of mobile applications. Their checklist, Acc-MobileCheck, contains 47 items that correspond to issues related to comprehension (C), operation (O), perception (P), and adaptation (A) in mobile interface interaction. To validate Acc-MobileCheck, it was reviewed by five experts and three developers and determined to be effective. In *Pereda, Murillo & Paz (2020)*, the authors also suggest a set of heuristics to evaluate the accessibility of mobile e-commerce applications for visually impaired people. Finally, *McKay (2017)* conducted an accessibility test for hybrid mobile apps and found that students with blindness faced many barriers to access based on how they used hybrid mobile applications. While hybrid apps can allow for increased time for marketing, this comes at the cost of app accessibility for people with disabilities.

## B. Assistive devices

A significant number of people with visual impairments use state-of-the-art software to perform tasks in their daily lives. These technologies are made up of electronic devices equipped with sensors and processors that can make intelligent decisions.

One of the most important and challenging tasks in developing such technologies is to create a user interface that is appropriate for the sensorimotor capabilities of users with blindness (*Csapó et al., 2015*). Several new hardware tools have proposed to improve the quality of life for people with visual impairments. Three tools were presented in this SLR: a smart stick that can notify the user of any obstacle, helping them to perform tasks easily and efficiently (*Bharatia, Ambawane & Rane, 2019*), and an eye that can allow users to detect colors (medical evaluation is still required) (*Lewis et al., 2016*).

The purpose of the study in *Griffin-Shirley et al. (2017)* was to understand how people with blindness use smartphone applications as assistive technology and how they perceive them in terms of accessibility and usability. An online survey with 259 participants was conducted, and most of the participants rated the applications as useful and accessible and were satisfied with them.

The researchers in *Rahman, Anam & Yeasin (2017)* designed and implemented EmoAssist, which is a smartphone application that assists with natural dyadic conversations and aims to promote user satisfaction by providing options for accessing non-verbal communication that predicts behavioural expressions and contains interactive dimensions to provide valid feedback. The usability of this application was evaluated in a study with ten people with blindness where several tools were applied in the application. The study participants found that the usability of EmoAssist was good, and it was an effective assistive solution.

### C. Daily activities

This theme contains two main categories: braille-based application studies and applications to enhance the independence of VIU. Both are summarized below.

#### 1- Braille-based applications

Braille is still the most popular method for assisting people with visual impairments in reading and studying, and most educational mobile phone applications are limited to sighted people. Recently, however, some researchers have developed assistive education applications for students with visual impairments, especially those in developing countries. For example, in India, the number of children with visual impairments is around 15 million, and only 5% receive an education (*Gokhale et al., 2017*). Three of the braille studies focused on education: (*Nahar, Sulaiman & Jaafar, 2020*; *Nahar, Jaafar & Sulaiman, 2019*, and *Araújo et al., 2016*). These studies all used smartphone touchscreens and action gestures to gain input from the student, and then output was provided in the form of audio feedback. In *Nahar, Sulaiman & Jaafar (2020)*, vibrational feedback was added to guide the users. The participants in *Nahar, Sulaiman & Jaafar (2020)*; *Nahar, Jaafar & Sulaiman (2019)*, and *Araújo et al. (2016)* included students with blindness of visual impairment and their teachers. The authors in *Nahar, Sulaiman & Jaafar (2020)*, *Nahar, Jaafar & Sulaiman (2019)* evaluated the usability of their applications following the same criteria (efficiency, learnability, memorability, errors, and satisfaction). The results showed that in *Nahar, Sulaiman & Jaafar (2020)*, *Nahar, Jaafar & Sulaiman (2019)*, and *Araújo et al. (2016)*, the applications met the required usability criteria. The authors in *Gokhale et al. (2017)* presented a braille-based solution to help people with visual impairments call and save contacts. A braille keypad on the smartphone touchscreen was used to gain input from the user, which was then converted into haptic and auditory feedback to let the user know what action was taken. The usability of this application was considered before it was designed. The participants' responses were positive because this kind of user-centric design simplifies navigation and learning processes.

#### 2- Applications to enhance the independence of people with visual impairments

The authors in the studies explored in this section focused on building applications that enhance independence and autonomy in daily life activities for users with visual impairments.

In *Vitiello et al. (2018)*, the authors presented their mobile application, an assistive solution for visually impaired users called "Crania", which uses machine learning techniques to help users with visual impairments get dressed by recognizing the colour and texture of their clothing and suggesting suitable combinations. The system provides feedback through voice synthesis. The participants in the study were adults and elderly people, some of whom were completely blind and the rest of whom had partial sight. After testing for usability, all the participants with blindness agreed that using the application was better than their original method, and half of the participants with partial sight said the same thing. At the end of the study, the application was determined to be accessible and easy to use.

In *Kunaratana-Angkul, Wu & Shin-Renn (2020)*, an application which allows elderly people to measure low vision status at home through their smartphones instead of visiting hospitals was tested, and most of the participants considered it to be untrustworthy because the medical information was insufficient. Even when participants were able to learn how to use the application, most of them were still confused while using it and needed further instruction.

In *Ghidini et al. (2016)*, the authors studied the habits of people with visual impairments when using their smartphones in order to develop an electronic calendar with different interaction formats, such as voice commands, touch, and vibration interaction. The authors presented the lessons learned and categorized them based on usability heuristics such as feedback, design, user freedom and control, and recognition instead of remembering.

In *Madrigal-Cadavid et al. (2019)*, the authors developed a drug information application for people with visual impairments to help them access the labels of medications. The application was developed based on a user-centered design process. By conducting a usability test, the authors recognized some usability issues for people with visual impairments, such as difficulty in locating the bar code. Given this, a new version will include a search function that is based on pictures. The application is searched by capturing the bar code or text or giving voice commands that allow the user to access medication information. The participants were people with visual impairments, and most of them required assistance using medications before using the application. This application will enhance independence for people with visual impairments in terms of using medications.

In *Marques, Carriço & Guerreiro (2015)*, an authentication method is proposed for users with visual impairments that allows them to protect their passwords. It is not secure when blind or visually impaired users spell out their passwords or enter the numbers in front of others, and the proposed solution allows the users to enter their password with one hand by tapping the screen. The blind participants in this study demonstrated that this authentication method is usable and supports their security needs.

In *Oliveira et al. (2018)*, the author noted that people with visual impairments face challenges in reading, thus he proposed an application called LeR otulos. This application was developed and evaluated for the Android operating system and recognizes text from photos taken by the mobile camera and converts them into an audio description. The prototype was designed to follow the guidelines and recommendations of usability and accessibility. The requirements of the application are defined based on the following usability goals: the steps are easy for the user to remember; the application is efficient, safe, useful, and accessible; and user satisfaction is achieved.

Interacting with talkback audio devices is still difficult for people with blindness, and it is unclear how much benefit they provide to people with visual impairments in their daily activities. The author in *Rodrigues et al. (2015)* investigates the smartphone adoption process of blind users by conducting experiments, observations, and weekly interviews. An eight-week study was conducted with five visually impaired participants using Samsung and an enabled talkback 2 screen reader. Focusing on understanding the experiences of people with visual impairments when using touchscreen smartphones revealed accessibility and usability issues. The results showed that the participants have difficulties using smartphones

because they fear that they cannot use them properly, and that impacts their ability to communicate with family. However, they appreciate the benefits of using smartphones in their daily activities, and they have the ability to use them.

## D. Screen division layout

People with visual impairments encounter various challenges identifying and locating non-visual items on touch screen interfaces, such as phones and tablets. Various specifications for developing a user interface for people with visual impairments must be met, such as having touch screen division to enable people with blindness to easily and comfortably locate objects and items that are non-visual on the screen (*Khusro et al., 2019*). Article (*Khusro et al., 2019*) highlighted the importance of aspects of the usability analysis, such as screen partitioning, to meet specific usability requirements, including orientation, consistency, operation, time consumption, and navigation complexity when users want to locate objects on their touchscreen. The authors of *Khan & Khusro (2019)* describe the improvements that people with blindness have experienced in using the smartphone while performing their daily tasks. This information was determined through an empirical study with 41 people with blindness who explained their user and interaction experiences operating a smartphone.

The authors in *Palani et al. (2018)* provide design guidelines governing the accurate display of haptically perceived graphical materials. Determining the usability parameters and the various cognitive abilities required for optimum and accurate use of device interfaces is crucial. Also the authors of *Grussenmeyer & Folmer (2017)* highlight the importance of usability and accessibility of smartphones and touch screens for people with visual impairments. The primary focus in *Leporini & Palmucci (2018)* is on interactive tasks used to finish exercises and to answer questionnaires or quizzes. These tools are used for evaluation tests or in games. When using gestures and screen readers to interact on a mobile device, difficulties may arise (*Leporini & Palmucci, 2018*), The study has various objectives, including gathering information on the difficulties encountered by people with blindness during interactions with mobile touch screen devices to answer questions and investigating practicable solutions to solve the detected accessibility and usability issues. A mobile app with an educational game was used to apply the proposed approach. Moreover, in *Alnfiai & Sampalli (2016)* and *Niazi et al. (2016)*, an analysis of the single-tap braille keyboard created to help people with no or low vision while using touch screen smartphones was conducted. The technology used in *Alnfiai & Sampalli (2016)* was the talkback service, which provides the user with verbal feedback from the application, allowing users with blindness to key in characters according to braille patterns. To evaluate single tap braille, it was compared to the commonly used QWERTY keyboard. In *Niazi et al. (2016)*, it was found that participants adapted quickly to single-tap Braille and were able to type on the touch screen within 15 to 20 min of being introduced to this system. The main advantage of single tap braille is that it allows users with blindness to enter letters based on braille coding, which they are already familiar with. The average error rate is lower using single-tap Braille than it is on the QWERTY keyboard. The authors of *Niazi et al. (2016)* found that minimal typing errors were made using the proposed keypad, which made it an easier option for

people with blindness (*Niazi et al., 2016*). In *Cho & Kim (2017)*, the authors describe new text entry methods for the braille system including a left touch and a double touch scheme that form a two-button interface for braille input so that people with visual impairments are able to type textual characters without having to move their fingers to locate the target buttons.

## E. Gestures

One of the main problems affecting the visually impaired is limited mobility for some gestures. We need to know what gestures are usable by people with visual impairments. Moreover, the technology of assistive touchscreen-enabled speech interaction between blind people and mobile devices permits the use of gestures to interact with a touch user interface. Assistive technology is vital in helping people living with disabilities to perform actions or interact with systems. *Smaradottir, Martinez & Haland (2017)* analyses a voiceover screen reader used in Apple Inc.'s products. An assessment of this assistive technology was conducted with six visually impaired test participants. The main objectives were to pinpoint the difficulties related to the performance of gestures applicable in screen interactions and to analyze the system's response to the gestures. In this study, a user evaluation was completed in three phases. The first phase entailed training users regarding different hand gestures, the second phase was carried out in a usability laboratory where participants were familiarized with technological devices, and the third phase required participants to solve different tasks. In *Knutas et al. (2015)*, the vital feature of the system is that it enables the user to interactively select a 3D scene region for sonification by merely touching the phone screen. It uses three different modes to increase usability. *Alnfiai & Sampalli (2017)* explained a study done to compare the use of two data input methods to evaluate their efficiency with completely blind participants who had prior knowledge of braille. The comparison was made between the braille enter input method that uses gestures and the swift braille keyboard, which necessitates finding six buttons representing braille dots. Blind people typically prefer rounded shapes to angular ones when performing complex gestures, as they experience difficulties performing straight gestures with right angles. Participants highlighted that they experienced difficulties particularly with gestures that have steep or right angles. In *Buzzi et al. (2017)*, 36 visually impaired participants were selected and split into two groups of low-vision and blind people. They examined their touch-based gesture preferences in terms of the number of strokes, multitouch, and shape angles. For this reason, a wireless system was created to record sample gestures from various participants simultaneously while monitoring the capture process.

## F. Audio guidance

People with visual impairment typically cannot travel without guidance due to the inaccuracy of current navigation systems in describing roads and especially sidewalks. Thus, the author of *Gintner et al. (2017)* aims to design a system to guide people with visual impairments based on geographical features and addresses them through a user interface that converts text to audio using a built-in voiceover engine (Apple iOS). The system was evaluated positively in terms of accessibility and usability as tested in a qualitative study involving six participants with visual impairment.

Based on challenges faced by visually impaired game developers, *Ara'ujo et al. (2017)* provides guidance for developers to provide accessibility in digital games by using audio guidance for players with visual impairments. The interactions of the player can be conveyed through audio and other basic mobile device components with criteria focused on the game level and speed adjustments, high contrast interfaces, accessible menus, and friendly design. Without braille, people with visual impairments cannot read, but braille is expensive and takes effort, and so it is important to propose technology to facilitate reading for them. In *Sabab & Ashmafee (2016)*, the author proposed developing a mobile application called "Blind Reader" that reads an audio document and allows the user to interact with the application to gain knowledge. This application was evaluated with 11 participants, and the participants were satisfied with the application. Videos are an important form of digital media, and unfortunately people with visual impairment cannot access these videos. Therefore, *Façanha et al. (2016)* aims to discover sound synthesis techniques to maximize and accelerate the production of audio descriptions with low-cost phonetic description tools. This tool has been evaluated based on usability with eight people and resulted in a high acceptance rate among users.

## G. Navigation

### 1- Indoor navigation

Visually impaired people face critical problems when navigating from one place to another. Whether indoors or outdoors, they tend to stay in one place to avoid the risk of injury or seek the help of a sighted person before moving (*Al-Khalifa & Al-Razgan, 2016*). Thus, aid in navigation is essential for those individuals. In *Nair et al. (2020)*, Nair developed an application called ASSIST, which leverages Bluetooth low energy (BLE) beacons and augmented reality (AR) to help visually impaired people move around cluttered indoor places (*e.g.*, subways) and provide the needed safe guidance, just like having a sighted person lead the way. In the subway example, the beacons will be distributed across the halls of the subway and the application will detect them. Sensors and cameras attached to the individual will detect their exact location and send the data to the application. The application will then give a sequence of audio feedback explaining how to move around the place to reach a specific point (*e.g.*, "in 50 ft turn right", "now turn left", "you will reach the destination in 20 steps"). The application also has an interface for sighted and low-vision users that shows the next steps and instructions. A usability study was conducted to test different aspects of the proposed solution. The majority of the participants agreed that they could easily reach a specified location using the application without the help of a sighted person. A survey conducted to give suggestions from the participants for future improvements showed that most participants wanted to attach their phones to their bodies and for the application to consider the different walking speeds of users. They were happy with the audio and vibration feedback that was given before each step or turn they had to take.

In *Al-Khalifa & Al-Razgan (2016)*, the main purpose of the study was to provide an Arabic-language application for guidance inside buildings using Google Glass and an

associated mobile application. First, the building plan must be set by a sighted person who configures the different locations needed. Ebsar will ask the map builder to mark each interesting location with a QR code and generate a room number, and the required steps and turns are tracked using the mobile device's built-in compass and accelerometer features. All of these are recorded in the application for the use of a visually impaired individual, and at the end, a full map is generated for the building. After setting the building map, a user can navigate inside the building with the help of Ebsar, paired with Google Glass, for input and output purposes. The efficiency, effectiveness, and levels of user satisfaction with this solution were evaluated. The results showed that the errors made were few, indicating that Ebsar is highly effective. The time consumed in performing tasks ranged from medium to low depending on the task; this can be improved later. Interviews with participants indicated the application's ease of use. *De Borba Campos et al. (2015)* shows an application simulating a museum map for people with visual impairments. It discusses whether mental maps and interactive games can be used by people with visual impairments to recognize the space around them. After multiple usability evaluation sessions, the mobile application showed high efficiency among participants in understanding the museum's map without repeating the visitation. The authors make a few suggestions based on feedback from the participants regarding enhancing usability, including using audio cues, adding contextual help to realise the activities carried around in a space, and focusing on audio feedback instead of graphics.

### 2- Outdoor navigation

Outdoor navigation is also commonly discussed in the literature. In *Darvishy et al. (2020)*, Touch Explorer was presented to alleviate many of the problems faced by visually impaired people in navigation by developing a non-visual mobile digital map. The application relies on three major methods of communication with the user: voice output, vibration feedback, and everyday sounds. The prototype was developed using simple abstract visuals and mostly relies on voice for explanation of the content. Usability tests show the great impact the prototype had on the understanding of the elements of the map. Few suggestions were given by the participants to increase usability, including GPS localization to locate the user on the map, a scale element for measuring the distance between two map elements, and an address search function.

In *Hossain, Qaiduzzaman & Rahman (2020)*, a navigation application called Sightless Helper was developed to provide a safe navigation method for people with visual impairments. It relies on footstep counting and GPS location to provide the needed guidance. It can also ensure safe navigation by detect objects and unsafe areas and can detect unusual shaking of the user and alert an emergency contact about the problem. The user interaction categories are voice recognition, touchpad, buttons, and shaking sensors. After multiple evaluations, the application was found to be useful in different scenarios and was considered usable by people with visual impairments. The authors in *Long et al. (2016)* propose an application that uses both updates from users and information about the real world to help visually impaired people navigate outdoor settings. After interviews with participants, some design goals were set, including the ability to tag an obstacle

on the map, check the weather, and provide an emergency service. The application was evaluated and was found to be of great benefit; users made few errors and found it easy to use. In *Prerana et al. (2019)*, a mobile application called STAVI was presented to help visually impaired people navigate from a source to a destination safely and avoid issues of re-routing. The application depends on voice commands and voice output. The application also has additional features, such as calling, messages, and emergency help. The authors in *Bandukda et al. (2020)* helped people with visual impairments explore parks and natural spaces using a framework called PLACES. Different interviews and surveys were conducted to identify the issues visually impaired people face when they want to do any leisure activity. These were considered in the development of the framework, and some design directions were presented, such as the use of audio to share an experience.

### 3- General issues

The authors in *Maly et al. (2015)* discuss implementing an evaluation model to assess the usability of a navigation application and to understand the issues of communication with mobile applications that people with visual impairments face. The evaluation tool was designed using a client–server architecture and was applied to test the usability of an existing navigation application. The tool was successful in capturing many issues related to navigation and user behavior, especially the issue of different timing between the actual voice instruction and the position of the user. The authors in *Kameswaran et al. (2020)* conducted a study to find out which navigation technologies blind people can use and to understand the complementarity between navigation technologies and their impact on navigation for visually impaired users. The results of the study show that visually impaired people use both assistive technologies and those designed for non-visually impaired users. Improving voice agents in navigation applications was discussed as a design implication for the visually impaired. In *Skulimowski et al. (2019)*, the authors show how interactive sonification can be used in simple travel aids for the blind. It uses depth images and a histogram called U-depth, which is simple auditory representations for blind users. The vital feature of this system is that it enables the user to interactively select a 3D scene region for sonification by touching the phone screen. This sonic representation of 3D scenes allows users to identify the environment's general appearance and determine objects' distance. The prototype structure was tested by three blind individuals who successfully performed the indoor task. Among the test scenes used included walking along an empty corridor, walking along a corridor with obstacles, and locating an opening between obstacles. However, the results showed that it took a long time for the testers to locate narrow spaces between obstacles.

### RQ6: What evaluation methods were used in the studies on usability for visually impaired people that were reviewed?

The most prevalent methods to evaluate the usability of applications were surveys and interviews. These were used to determine the usability of the proposed solutions and obtain feedback and suggestions regarding additional features needed to enhance the usability from the participants' points of view. Focus groups were also used extensively in the literature. Many of the participants selected were blindfolded and were not actually

blind or visually impaired. Moreover, the samples selected for the evaluation methods mentioned above considered the age factor depending on the study's needs.

## LIMITATION AND FUTURE WORK

The limitations of this paper are mainly related to the methodology followed. Focusing on just eight online databases and restricting the search with the previously specified keywords and string may have limited the number of search results. Additionally, a large number of papers were excluded because they were written in other languages. Access limitations were also faced due to some libraries asking for fees to access the papers. Therefore, for future works, a study to expand on the SLR results and reveal the current usability models of mobile applications for the visually impaired to verify the SLR results is needed so that this work contributes positively to assessing difficulties and expanding the field of usability of mobile applications for users with visual impairments.

## CONCLUSIONS

In recent years, the number of applications focused on people with visual impairments has grown, which has led to positive enhancements in those people's lives, especially if they do not have people around to assist them. In this paper, the research papers focusing on usability for visually impaired users were analyzed and classified into seven themes: accessibility, daily activities, assistive devices, gestures, navigation, screen division layout, and audio guidance. We found that various research studies focus on accessibility of mobile applications to ensure that the same user experience is available to all users, regardless of their abilities. We found many studies that focus on how the design of the applications can assist in performing daily life activities like braille-based application' studies and applications to enhance the independence of VI users. We also found papers that discuss the role of assistive devices like screen readers and wearable devices in solving challenges faced by VI users and thus improving their quality of life. We also found that some research papers discuss limited mobility of some gestures for VI users and investigated ways in which we can know what gestures are usable by people with visual impairments. We found many research papers that focus on improving navigation for VI users by incorporating different output modalities like sound and vibration. We also found various studies focusing on screen division layout. By dividing the screen and focusing on visual impairment-related issues while developing user interfaces, visually impaired users can easily locate the objects and items on the screens. Finally, we found papers that focus on audio guidance to improve usability. The proposed applications use voice-over and speech interactions to guide visually impaired users in performing different activities through their mobiles. Most of the researchers focused on usability in different applications and evaluated the usability issues of these applications with visually impaired participants. Some of the studies included sighted participants to compare the number and type of problems they faced. The usability evaluation was generally based on the following criteria: accessibility, efficiency, learnability, memorability, errors, safety, and satisfaction. Many of the studied applications show a good indication of these applications' usability and

follow the participants' comments to ensure additional enhancements in usability. This paper aims to provide an overview of the developments on usability of mobile applications for people with visual impairments and use this overview to highlight potential future directions.

## APPENDIX

References selected for each theme.

| No. | Name of the study | Category |
|---|---|---|
| **Table A1** | **References selected for each theme.** | |
| 1. | Making mobile map applications accessible for visually impaired people | Accessibility |
| 2. | With most of it being pictures now, I rarely use it' Understanding Twitter's Evolving Accessibility to Blind Users. | Accessibility |
| 3. | Usability of user-centric mobile application design from visually impaired people's perspective. | Accessibility |
| 4. | Blindsense: An accessibility-inclusive universal user interface for blind people. | Accessibility |
| 5. | Acc-MobileCheck: a Checklist for Usability and Accessibility Evaluation of Mobile Applications. | Accessibility |
| 6. | Visually Impaired Accessibility Heuristics Proposal for e-Commerce Mobile Applications. | Accessibility |
| 7. | Accessibility or Usability of the User Interfaces for Visually Impaired Users? A Comparative Study. | Accessibility |
| 8. | Accessibility and Usability Problems Encountered on Websites and Applications in Mobile Devices by Blind and Normal-Vision Users. | Accessibility |
| 9. | Preliminary Evaluation of Interactive Search Engine Interface for Visually Impaired Users. | Accessibility |
| 10. | Accessibility Challenges of Hybrid Mobile Applications. | Accessibility |
| 11. | Evaluation of touchscreen assistive technology for visually disabled users. | Assistive devices |
| 12. | Interactive sonification of U-depth images in a navigation aid for the visually impaired. | Assistive devices |
| 13. | UniPass: design and evaluation of a smart device-based password manager for visually impaired users. | Assistive devices |
| 14. | The Effect of the Usage of Computer-Based Assistive Devices on the Functioning and Quality of Life of Individuals who are Blind or have low Vision. | Assistive devices |
| 15. | A survey of assistive technologies and applications for blind users on mobile platforms: a review and foundation for research. | Assistive devices |

**Table A1** (*continued*)

| No. | Name of the study | Category |
|---|---|---|
| 16. | An insight into smartphone-based assistive solutions for visually impaired and blind people: issues, challenges and opportunities. | Assistive devices |
| 17. | OCR based facilitator for the visually challenged. | Assistive devices |
| 18. | The interaction experiences of visually impaired people with assistive technology: A case study of smartphones. | Assistive devices |
| 19. | Social Media Platforms for Low-Income Blind People in India. | Assistive devices |
| 20. | Understanding In-Situ Use of Commonly Available Navigation Technologies by People with Visual Impairments. | Assistive devices |
| 21. | A Survey on the Use of Mobile Applications for People who Are Visually Impaired. | Assistive devices |
| 22. | Smart Electronic Stick for Visually Impaired using Android Application and Google's Cloud Vision. | Assistive devices |
| 23. | Advances in implantable bionic devices for blindness: a review. | Assistive devices |
| 24. | An interactive math braille learning application to assist blind students in Bangladesh. | Daily activities |
| 25. | Usability evaluation of a mobile phone-based braille learning application 'mbraille. | Daily activities |
| 26. | Design and usability of a braille-based mobile audiogame environment. | Daily activities |
| 27. | SparshJa: A User-Centric Mobile Application Designed for Visually Impaired. | Daily activities |
| 28. | Do you like my outfit?: Cromnia, a mobile assistant for blind users. | Daily activities |
| 29. | Usability in the app Interface Designing for the Elderly with Low-Vision in Taiwan and Thailand. | Daily activities |
| 30. | Developing Apps for Visually Impaired People: Lessons Learned from Practice. | Daily activities |
| 31. | Design and development of a mobile app of drug information for people with visual impairment. | Daily activities |
| 32. | Assessing Inconspicuous Smartphone Authentication for Blind People. | Daily activities |
| 33. | LR$^s$otulos: A Mobile Application Based on Text Recognition in Images to Assist Visually Impaired People. | Daily activities |
| 34. | Getting Smartphones to Talkback: Understanding the Smartphone Adoption Process of Blind Users. | Daily activities |
| 35. | Evaluating Smartphone Screen Divisions for Designing Blind-Friendly Touch-Based Interfaces. | Screen division layout |
| 36. | Blind-friendly user interfaces–a pilot study on improving the accessibility of touchscreen interfaces. | Screen division layout |
| 37. | Accessible touchscreen technology for people with visual impairments: a survey. | Screen division layout |
| 38. | Touchscreen-based haptic information access for assisting blind and visually-impaired users: Perceptual parameters and design guidelines. | Screen division layout |
**Table A1** (*continued*)

| No. | Name of the study | Category |
|---|---|---|
| 39. | Accessible Question Types on a Touch-Screen Device: The Case of a Mobile Game App for Blind People. | Screen division layout |
| 40. | Touchscreen Based Text-Entry for Visually-Impaired Users. | Screen division layout |
| 41. | An evaluation of SingleTapBraille keyboard: a text entry method that utilizes braille patterns on touchscreen devices. | Screen division layout |
| 42. | A touch-sensitive keypad layout for improved usability of smartphones for the blind and visually impaired persons. | Screen division layout |
| 43. | BraillePassword: accessible web authentication technique on touchscreen devices. | Screen division layout |
| 44. | An evaluation of the brailleenter keyboard: An input method based on braille patterns for touchscreen devices. | Gestures |
| 45. | Analyzing visually impaired people's touch gestures on smartphones. | Gestures |
| 46. | Evaluation of touchscreen assistive technology for visually disabled users. | Gestures |
| 47. | Improving reverse geocoding: Localization of blind pedestrians using conversational UI. | Audio guidance |
| 48. | Mobile Audio Games Accessibility Evaluation for Users Who Are Blind. | Audio guidance |
| 49. | Blind Reader: An intelligent assistant for blind. | Audio guidance |
| 50. | Audio Description of Videos for People with Visual Disabilities. | Audio guidance |
| 51. | Ebsar: Indoor guidance for the visually impaired," Computers & Electrical Engineering. | Navigation |
| 52. | ASSIST: Evaluating the usability and performance of an indoor navigation assistant for blind and visually impaired people. | Navigation |
| 53. | Usability evaluation of a mobile navigation application for blind users. | Navigation |
| 54. | Touch Explorer: Exploring Digital Maps for Visually Impaired People. | Navigation |
| 55. | Emotion enabled assistive tool to enhance dyadic conversation for the blind | Navigation |
| 56. | Sightless Helper: An Interactive Mobile Application for Blind Assistance and Safe Navigation. | Navigation |
| 57. | Using a mobile application to help visually impaired individuals explore the outdoors. | Navigation |
| 58. | STAVI: Smart Travelling Application for the Visually Impaired. | Navigation |
| 59. | PLACES: A Framework for Supporting Blind and Partially Sighted People in Outdoor Leisure Activities. | Navigation |
| 60. | Qualitative measures for evaluation of navigation applications for visually impaired. | Navigation |

### Funding

This research project was supported by a grant from the Research Center of the Female Scientific and Medical Colleges, Deanship of Scientific Research, King Saud University. The funders had no role in study design, data collection and analysis, decision to publish, or preparation of the manuscript.

### Grant Disclosures

The following grant information was disclosed by the authors:
The Research Center of the Female Scientific and Medical Colleges, Deanship of Scientific Research, King Saud University.

### Competing Interests

The authors declare there are no competing interests.

### Author Contributions

- Muna Al-Razgan, Sarah Almoaiqel, Nuha Alrajhi, Alyah Alhumegani, Abeer Alshehri, Bashayr Alnefaie, Raghad AlKhamiss and Shahad Rushdi conceived and designed the experiments, performed the experiments, analyzed the data, performed the computation work, prepared figures and/or tables, authored or reviewed drafts of the paper, and approved the final draft.

### Data Availability

  This is a systematic literature review; there is no raw data.

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
