# Peer review of "A systematic literature review on the usability of mobile applications for visually impaired users"

_PeerJ Computer Science, doi:10.7717/peerj-cs.771_

## Round 0.1 · original submission · Major Revisions

Please see all three reviewers comments. I hope to receive the revised manuscript addressing all the required minor and major changes.

Reviewer 1 ·

Basic reporting

The review is of interest and within the scope of the journal. The documents presents a A Systematic Literature Review on the Usability of Mobile Applications for Visually Impaired Users. I know of a recent review about this topic but the document is in Spanish and is specifically about indoor technologies (View publication at https://latamt.ieeer9.org/index.php/transactions/article/view/4454). I suggest improving the abstract at lines 23- 24 to provide more justification for your study (specifically, you should expand upon the knowledge gap being filled).

Other additional comments are:

How does this figure (line 63) illustrate the classification? You are repeating the same classification information you already mentioned in here and added a % to each theme.

What do you mean by "concludes the paper" in line 68. I mean, What is "that section: that concludes the paper?

Is this figure in line 85 really necessary? You already mentioned the stages on the text. Besides, you don't mention the figure. Figures should not be placed without having been mentioned in the text.

What do you mean by relevant? This is also mentioned in Table 2 but you need to be more specific about this unique inclusion/exclusion criteria (lines 135, 157, 158).

Same comments/questions as before: Is this figure necessary? (Line 136) You already mentioned the number of papers per database in the text. Besides, you did not mention the figure. Figures should not be placed without having been mentioned in the text.

At line 138- 139 It should say sub stage. I suggest to be more specific. Please rewrite this sentence as: "Therefore, importing citations into MENDELEY was mandatory in order to eliminate the duplicates.

About figure 4: Same comment. This figure is not mentioned in the text.

Figure is not a summary of just the stage mentioned, is a summary of all stages, hence, it should be mentioned before, as indicated.

Figures should be placed the closest AFTER after mentioning them the first time.

Please rewrite lines 207 and 208 as: The PRISMA diagram shown in Figure 7 illustrates all systematic literature processes used in this study.

Be more specific in line 211. All researchers involved in this SLR...

Experimental design

The survey methodology is ver consistent and well explained. However, I perceived a language-biased coverage of the subject. Why just documents in English? You need to add some justification/explanation.

Validity of the findings

The paper illustrates the different trends, themes, and evaluation results of various mobile applications developed in the last six years. Using this overview as a foundation, future directions for research in the field of usability for the visually impaired (UVI) are highlighted. More discussion about those futures directions needs to be added.

Annotated reviews are not available for download in order to protect the identity of reviewers who chose to remain anonymous.

Reviewer 2 ·

Basic reporting

The paper presents a systematic literature review on the usability of mobile applications for visually impaired users, covering a synthesis of 60 studies selected from an initial sample of 932 studies encountered in the literature from the last six years.

My first suggestion for improvements is to include more details of the findings and type of knowledge synthesized in the study in the abstract. The information on results and their impact is very briefly described in the abstract, which would jeopardize the chances of reader having a broader view of what the paper has to offer before reading the full text.

There are very long paragraphs in the text. The first paragraph of the introduction is very long, and I would strongly recommend the authors break it down into shorter paragraphs.

The introduction has definitions to key terms, such as usability, with no reference to authoritative sources.

I also believe it is important that the authors highlight the research gap they identified and that motivated the present study. Have other studies performed systematic reviews on the usability of mobile applications for visually impaired people? Why did the authors choose the period of six years?

Line 52: fix citation style: evaluation tests. (Bastien, 2010).

Experimental design

The search protocol should describe the different adaptations the search string had to undergo to be executed in each database.

The quality assessment, starting on line 180, does not describe which authors participated in the procedures. It is important to describe how the process was conducted and, if there were disagreements, how they were solved.

Validity of the findings

The statement on line 227 is very confusing. What do authors mean by “Of a total of 60 studies, 10 discussed accessibility.”. Considering the topic of the study, how could they not consider accessibility?

In fact, further reading the paper reveals that the categories chosen to organize the studies has little justification and explanation in terms of conceptualization. There is very little depth in the description of the categories chosen for the papers.

There are several inconsistencies in the citation style that need to be fixed.

Unfortunately, the current version of the paper does not present enough consolidation of knowledge in the field, with superficial analyses with little conceptualization to provide a broader overview of the area.

In my view, the systematic review would need a new analysis, with a substantially deeper analysis and conceptualization.

Additional comments

no comments.

Reviewer 3 ·

Basic reporting

The study aims to find the usability issues for the visually impaired in mobile applications. The authors systematically searched and analyzed the papers on the usability of mobile applications for the visually impaired. The paper reads well.

The references used in the Introduction section (Brady et, al and Bastien et. Al.) are pretty old and should be updated. There are no similar literature review papers mentioned in this review. The description and the relevance of the problem should be extended.

The authors mention that they classified the studies into five different themes in the Introduction section, but they should also explain how they chose them. Also, there are seven themes in Figure 1. At this point of reading the paper it is not clear what the percentages mean in Figure 1 since this is explained a few pages below.
I suggest that this part is excluded from Intruduction section.

Experimental design

Was this review registered?
I like Table 1, where all research questions are listed and the motivation for them is explained.
I don't see the contribution of Figure 2 to better undestandability of the paper and I propose the authors to exclude it.
The NAILS project webpage should be cited.
»Keyword protocol« in actually not a protocol - it is a search string.
Please explain the difference (benefit) in results obtained using the initial and the »new« search string.
Table 2 is also not needed. Everything can be explained in one or two sentences.
The legend »Series 1« at Figure 4 is not needed. The databases on y-axis should be listed in decreasing order.
Figure 9 is not a PRISMA 2009 Flow Diagram. Please refere to the guidances described at http://www.prisma-statement.org/ or. directly at (https://www.google.com/url?sa=t&rct=j&q=&esrc=s&source=web&cd=&ved=2ahUKEwi0yrv8wqjyAhVImYsKHU5cAGQQFnoECAQQAQ&url=http%3A%2F%2Fprisma-statement.org%2Fdocuments%2FPRISMA%25202009%2520flow%2520diagram.pdf&usg=AOvVaw1qm3ududj_3lshDSZ29LLL)
You should mention that you folowed PRISMA 2009 guidances for SR in the begining of the methodology section. Also, mention when did the searcing through the databases took place.

Validity of the findings

I think the reader would benefit from the table of selected papers where the authors could maybee present the summarization of the results from Discussion section (or some of them…). I know that 60 papers is a lot (for the SLR), but the reader has to search for information on which papers were selected through the text now.
When describing the study which involved participants, please includ information on the sample size (like in line 458; the description of the next study (Rahman et. All) does not include the information on number of participants. This would also be important feature in the table with all selected studies.
RV6: Please describe in more details what was the range of sample sizes in the studies and be more specific in reporting the findings with adding the references on the papers.

Additional comments

The references should be listed in alphabetic order if the chosen citation style is used.

---

## Round 0.2 · Minor Revisions

Please see reviewer 3's comments. The authors should include the references selected for the SLR in a tabular format. The table should contain the references selected for each theme.
In addition, the paper should be proofread for typos, grammar and other citations issues.
Finally, follow the journal guidelines for paper structure.
All the best!

Reviewer 1 ·

Basic reporting

No comment

Experimental design

No comment

Validity of the findings

No comment

Additional comments

The authors made all the suggested revisions and I am satisfied with their work.

Reviewer 3 ·

Basic reporting

The authors have included almost all of my suggestions in the reviewed manuscript.
I still miss a table with a short description of the papers included in the SLR (60 papers). This table could be included in the appendix. Right now I can not even confirm that there were really 60 papers included in the SLR - I would have to extract unique references from the description of the results.

Experimental design

The study methodology is adequate and well described.

Validity of the findings

The goals are meet and the research questions have been answered and discussed. The future directions are stated.

---

## Round 0.3 · accepted · Accept

Please double check all the figures and tables and make sure they are readable.